🔓 | **Open Peer Review** | Antimicrobial Chemotherapy | Research Article

# Development and validation of a nomogram-based risk prediction model for carbapenem-resistant *Klebsiella pneumoniae* in hospitalized patients

Tingting Xu,[1] Yuxin Shi,[2] Xiongjing Cao,[2] Lijuan Xiong[1,2]

**ABSTRACT**  Carbapenem-resistant *Klebsiella pneumoniae* (CRKP) poses one of the major challenges in clinical anti-infective therapy worldwide. This retrospective cohort study at a tertiary general hospital in Wuhan aimed to identify risk factors for hospital-acquired CRKP infections among 1,113 patients. All participants were aged 18 years and above, and had confirmed positive cultures for KP isolated within 48 hours post-hospitalization. Independent risk factors were identified using LASSO logistic regression and incorporated into a predictive nomogram. The factors included in the nomogram were prior carbapenem exposure, prior β-lactams–β-lactamase inhibitor combination (BLBLI) exposure, prior intensive care unit (ICU) stay, and prior mechanical ventilation. The areas under the receiver operating characteristic curve (AUC) for the nomogram were 0.793 in the training group (70% of patients) and 0.788 in the validation group (30% of patients), demonstrating its discriminatory power and predictive accuracy. The *P* values for the Hosmer-Lemeshow test were 0.333 and 0.684, indicating good calibration. The clinical utility of the nomogram was further supported by decision curve analysis (DCA) and clinical impact curve (CIC), demonstrating its potential to guide clinical decision-making. Our retrospective analysis identified key risk factors for CRKP infection and developed a nomogram that could effectively predict CRKP infections in hospitalized patients. Although the single-center nature of this study limits generalizability, the nomogram provides a foundation for future prospective, multicenter validations.

**IMPORTANCE**  We established a nomogram scoring system that incorporates four key risk factors: prior carbapenem exposure, prior β-lactams–β-lactamase inhibitor combination (BLBLI) exposure, prior intensive care unit (ICU) stay, and prior mechanical ventilation. This nomogram demonstrated strong discriminatory power, excellent calibration, and significant clinical utility. This study highlights the critical risk factors associated with hospital-acquired carbapenem-resistant *Klebsiella pneumoniae* (CRKP) infections, providing valuable insights for clinicians to identify high-risk patients.

**KEYWORDS**  carbapenem resistance, *Klebsiella pneumoniae*, nomogram, risk factor

*K*lebsiella pneumoniae (KP) is a member of intestinal flora and an opportunistic pathogen capable of causing various types of infections, such as respiratory infections, bloodstream infections, and intra-abdominal infections (1). Additionally, KP also ranks among the most common bacteria causing hospital-acquired infections (2). In the early 1980s, KP was found to be generally resistant to β-lactam antibiotics, and carbapenems, which have a broader antimicrobial spectrum, became the main antibiotics used to treat KP (3–5). However, in recent years, it has been found that carbapenem resistance mechanisms have emerged, and the incidence of carbapenem-resistant KP (CRKP) is rapidly increasing (6). In 2018 and 2019, the World Health

**Peer Reviewer** Innocent Afeke, University of Health and Allied Sciences, Ho, Volta Region, Ghana

Address correspondence to Lijuan Xiong, lijuanxiong2016@126.com.

Tingting Xu, Yuxin Shi, and Xiongjing Cao contributed equally to this article. The author order was decided collaboratively based on each contributor's contribution to the research.

The authors declare no conflict of interest.

See the funding table on p. 9.

Organization (WHO) and the Centers for Disease Control and Prevention (CDC) prioritized antibiotic-resistant bacteria based on criteria including mortality, healthcare impact, resistance prevalence, and transmissibility, dividing them into critical, high, and medium priority tiers at the 33rd percentile threshold, which CRKP was categorized as a critical priority and an urgent threat to human health (7, 8). This categorization underscores the alarming rates of carbapenem resistance observed in KP, the species most affected within the Enterobacterales order. Notably, resistance rates have reached highs in certain regions, such as Greece at 67%, Iran at 65%, Russia at 64%, India at 57%, Saudi Arabia at 50%, Peru at 45%, Italy at 33%, Argentina at 26%, and Brazil at 24% (9, 10). According to the China Antimicrobial Surveillance Network (CHINET, http://www.chinets.com/), in the first half of 2023, the detected resistance rates of KP to meropenem and imipenem in China were 30% and 29%, respectively (11). Due to high prevalence and mortality rates, CRKP has become a major public health hazard worldwide (12).

Effective prevention and control of CRKP infections require accurate risk assessment tools. Although existing research has identified factors such as prolonged hospitalization, intensive care unit (ICU) stay, KP colonization, receiving surgery, indwelling devices, as well as prior antibiotic exposure as potential predictors of CRKP (1, 13–15), these findings are often derived from small-sample, single-center observational studies that have not been validated on a large scale and have not generated prognostic models. Building on this foundation, our study leverages a large retrospective cohort from a tertiary care hospital in China to develop and validate a predictive nomogram for CRKP infection risk. This nomogram, grounded in extensive patient data and clinical parameters, offers a significant advancement by enhancing the accuracy of risk prediction. Our research contributes to the field by providing a practical tool that can help clinicians better anticipate CRKP infections. This nomogram not only aids in refining infection prevention strategies but also guides personalized treatment plans.

## MATERIALS AND METHODS

### Patients and study design

This retrospective study was conducted on patients with nosocomial infections of KP between 1 January 2020 and 30 April 2023 at a tertiary general hospital in Wuhan. All included patients had a positive culture for KP bacteria detected after 48 hours of hospitalization. The exclusion criteria were (i) age <18 years, (ii) duration of hospitalization <3 days, and (iii) incomplete data. Eligible patients were randomized into training and validation cohorts in a 7:3 ratio. A nomogram for predicting the risk of CRKP infection was developed in the training cohort, whose predictive accuracy was validated in both the training and validation cohorts. This study was approved by the Research Ethics Committee of Union Hospital, Tongji Medical College, Huazhong University of Science and Technology (approval number: 2023-S0792).

### Clinical data collection

The demographics and clinical variables collected included age, sex, duration of hospitalization before infection, types of infection, comorbidities, previous healthcare interventions, indwelling devices, exposure to antibiotics within 90 days. The type of infection was related to the anatomical site where the first positive culture specimen of KP is located (e.g., urine corresponded to a urinary tract infection). Neutropenia was defined as a neutrophil count of less than 1,000 /µL. Long-term use of immunosuppressor was defined as oral immunosuppression for more than 2 weeks. Chemotherapy was defined as the administration of intravenous chemotherapeutic drugs during hospitalization.

## Microbiological studies

Pathogen species were identified using the VITEK-2 system (Biomerieux, France). The agar dilution method was applied to determine carbapenem susceptibility, whose results were interpreted following Clinical and Laboratory Standards Institute (CLSI) standards. CRKP was defined as resistance to meropenem or imipenem with a minimum inhibitory concentration (MIC) ≥4 µg/mL.

## Statistical analysis

Continuous variables are presented as median and interquartile range (IQR), while categorical variables are presented as frequency and percentage (%). Continuous variables were compared by Wilcoxon rank sum test, and categorical variables were compared by chi-square test or Fisher's exact test. In the training group, all variables were initially screened for predictors of CRKP infection by least absolute shrinkage and selection operator (LASSO) regression. LASSO analysis narrowed the regression coefficients of the variables to zero through the penalty coefficient of lambda. After excluding variables with zero regression coefficients, all other variables were included in the multivariate logistic regression analysis using the step-forward (Wald) method to develop a predictive model for CRKP infection. The scores for each factor in the model were calculated and visualized with a nomogram. In the training and validation groups, receiver operating characteristic (ROC) curves and area under the ROC (AUC) were used to assess the model discrimination. The calibration curves were used to evaluate the prediction accuracy. The Hosmer-Lemeshow (H-L) test was used to assess the goodness of fit between predicted and observed probabilities ($P > 0.05$ for good calibration), and calibration with 1,000 bootstrap samples to decrease the overfit bias. Additionally, decision curve analysis (DCA) and clinical impact curves (CIC) were used to evaluate the effective benefit of the model in clinical practice. In all analyses, $P < 0.05$ was considered to indicate statistical significance. Statistical analysis was performed using the SPSS 26.0 software (IBM Corp., Armonk, NY, USA) and R version 4.1.2.

## RESULTS

### Clinical and demographic characteristics

A total of 1,113 patients with hospital-acquired KP infection were finally included in this study. All enrolled patients were randomized into a training group ($n = 779$) and a validation group ($n = 334$). The percentage of CRKP in the training and validation groups was 42.4% and 43.4%, respectively. There were no significant differences in clinical variables between the training and validation groups. In the training group, for example, the clinical characteristics included a median age (IQR) of 59 (49–68) years and a predominance of male patients (68.9%). The median duration of hospitalization (IQR) before KP infection was 12 (7–20) days. The most common type of infection was respiratory infection, followed by bloodstream infection. When positive KP was detected, 48.7% of the patients were hospitalized in or moved from the ICU. A total of 431 patients (55.3%) underwent surgery during hospitalization. Other major invasive procedures included tracheal intubation in 260 cases (33.4%), mechanical ventilation in 486 cases (62.4%), and urinary catheter intubation in 649 cases (83.3%), and 552 cases (70.9%) that received central venous catheterization.

### Risk factors of CRKP infection

All variables in Table 1 were preliminary screened using LASSO regression in the training group with CRKP infection as the dependent variable, and found that prior ICU stay, tracheal intubation, mechanical ventilation, exposure to carbapenems, and exposure to β-lactams–β-lactamase inhibitor combinations (BLBLIs) were significantly correlated with CRKP infection (Fig. 1). The four factors listed above were included in a multivariate logistic regression analysis, and the results revealed that prior carbapenem exposure (OR:

**TABLE 1** Participant characteristics of training and validation cohorts[a]

| Variable | Cohort, no. (%) | | P value |
|---|---|---|---|
| | Training (*n* = 779) | Validation (*n* = 334) | |
| Age, mean (IQR), y | 59 (49–68) | 58 (50–68) | 0.778 |
| Sex, male | 537 (68.9) | 235 (70.4) | 0.637 |
| Duration of hospitalization before infection, mean (IQR), d | 12 (7–20) | 12 (7–21) | 0.854 |
| Types of infection | | | 0.134 |
| Respiratory infection | 537 (68.9) | 235 (70.4) | |
| Bloodstream infection | 108 (13.9) | 37 (11.1) | |
| Genitourinary tract infection | 33 (4.2) | 13 (3.9) | |
| Intra-abdominal infection | 37 (4.8) | 27 (8.1) | |
| Urinary tract infection | 19 (2.4) | 9 (2.7) | |
| Central nervous system infection | 4 (0.5) | 2 (0.6) | |
| Skin and soft tissue infection | 41 (5.3) | 10 (3.0) | |
| Bone and joint infection | 0 (0.0) | 1 (0.3) | |
| Comorbidity | | | |
| Malignancy | 205 (26.3) | 87 (26.0) | 0.926 |
| Hypertension | 203 (26.1) | 96 (28.7) | 0.355 |
| Diabetes | 108 (13.9) | 51 (15.3) | 0.539 |
| Neutropenia | 87 (11.2) | 31 (9.3) | 0.349 |
| COPD | 49 (6.3) | 12 (3.6) | 0.070 |
| Fungal | 308 (39.5) | 126 (37.7) | 0.570 |
| Other comorbid bacterial infections | 473 (60.7) | 200 (59.9) | 0.793 |
| Chemotherapy | 104 (13.4) | 45 (13.5) | 0.956 |
| History of transplantation | 39 (5.0) | 17 (5.1) | 0.953 |
| Long-term use of immunosuppressor | 43 (5.5) | 22 (6.6) | 0.487 |
| Prior ICU stay | 379 (48.7) | 168 (50.3) | 0.614 |
| Prior surgery | 431 (55.3) | 182 (54.5) | 0.797 |
| Prior indwelling devices | | | |
| Tracheal intubation | 260 (33.4) | 113 (33.8) | 0.883 |
| Tracheotomy | 71 (9.1) | 21 (6.3) | 0.117 |
| Mechanical ventilation | 486 (62.4) | 202 (60.5) | 0.548 |
| Central system indwelling tube | 552 (70.9) | 236 (70.7) | 0.946 |
| Urinary catheter | 649 (83.3) | 272 (81.4) | 0.448 |
| Thoracic drainage tube | 9 (1.2) | 3 (0.9) | 0.703 |
| Abdominal drainage tube | 88 (11.3) | 41 (12.3) | 0.640 |
| Biliary drainage tube | 15 (1.9) | 7 (2.1) | 0.852 |
| Exposure to antibiotics within 90 days | | | |
| Carbapenems | 193 (24.8) | 83 (24.9) | 0.979 |
| Cephalosporins | 318 (40.8) | 131 (39.2) | 0.618 |
| Penicillins | 36 (4.6) | 17 (5.1) | 0.737 |
| BLBLIs | 369 (47.4) | 157 (47.0) | 0.912 |
| Quinolones | 106 (13.6) | 34 (10.2) | 0.114 |
| Aminoglycosides | 39 (5.0) | 11 (3.3) | 0.206 |
| Glycopeptides | 76 (9.8) | 27 (8.1) | 0.378 |
| Tigecycline | 100 (12.8) | 46 (13.8) | 0.672 |
| Linezolid | 20 (2.6) | 7 (2.1) | 0.639 |

[a]IQR, interquartile range; COPD, chronic obstructive pulmonary disease; ICU, intensive care unit; BLBLIs, β-lactams–β-lactamase inhibitor combinations; CRKP, carbapenem-resistant *K. pneumoniae*.

10.328, 95% CI: 6.740–15.825), prior BLBLI exposure (OR: 2.776, 95% CI: 1.982–3.889), prior ICU stay (OR: 1.472, 95% CI: 1.021–2.122), and prior mechanical ventilation (OR: 1.475, 95% CI: 1.002–2.172) were independent risk factors for CRKP infection (Table 2).

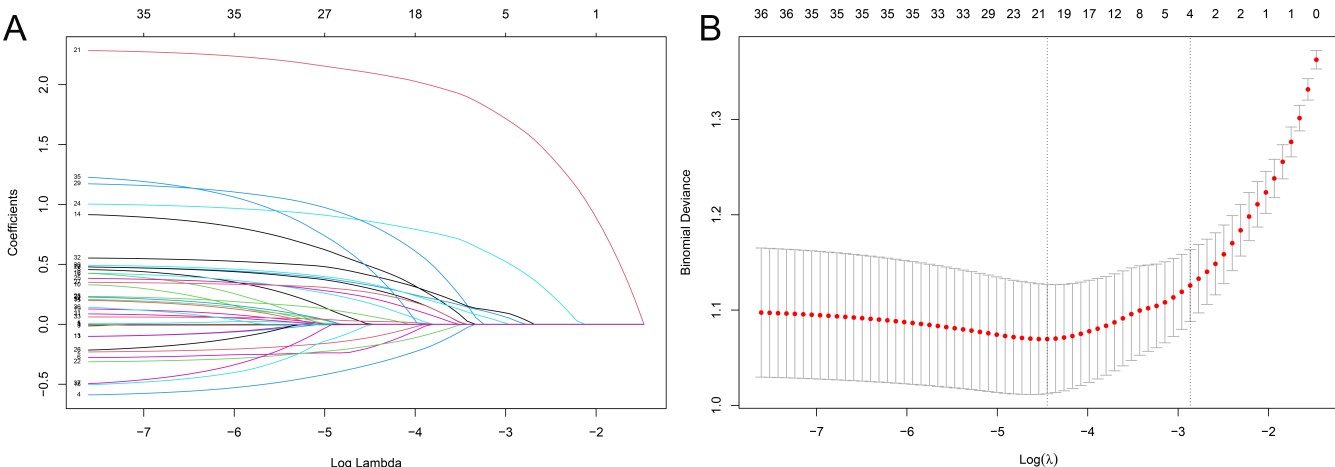

**FIG 1** (A) LASSO coefficient profiles of the 35 risk factors. Each curve in the figure presents the change of each variable in coefficient. (B) Tenfold cross-cross validation fitting and then selecting the mode.

## CRKP-predicting nomogram construction and validation

A predictive model for CRKP infection was developed with the four independent factors and presented by a nomogram, which showed that the probability of CRKP infection was more than 70% when the total score reached 112 and above (Fig. 2). As shown in Figure 3A, the ROC curves of both the training and validation cohorts demonstrated strong discrimination, with the optimal cutoff values being 0.448 and 0.400, respectively, and the AUC being 0.793 (95% CI: 0.760–0.825) and 0.788 (95% CI: 0.740–0.837), respectively. Hosmer-Lemeshow goodness of fit test presented $P$ values of 0.333 and 0.684 in the training cohort and validation cohort. Apparent calibration curves of the two cohorts also showed excellent agreement between the observed probability and predicted probability, with mean absolute errors of 0.017 and 0.014, respectively (Fig. 3B). All of the above indicates that the model was well calibrated. Furthermore, DCA and CIC both revealed that intervention decisions based on this model had evident clinical benefits (Fig. 3C and D).

## DISCUSSION

### Clinical relevance

Nosocomial infection is an important factor leading to prolonged hospital stays, aggravation of illness, and even fatalities in hospitalized patients (16, 17). KP, being the most prevalent bacterium responsible for nosocomial infections, necessitates concentrated, focused monitoring and control efforts (2). Additionally, KP is considered a crucial reservoir and vehicle for the dissemination of clinically significant antibiotic resistance genes (18). Consequently, gaining a deeper understanding of CRKP infections is imperative for mitigating the proliferation of antimicrobial resistance and guiding the appropriate application of antibiotics.

**TABLE 2** Multivariate logistic regression analysis of CRKP infection in the training cohort[a]

| Items | OR (95% CI) | $P$ value |
| --- | --- | --- |
| ICU stay | 1.472 (1.021–2.122) | 0.039 |
| Mechanical ventilation | 1.475 (1.002–2.172) | 0.049 |
| Exposure to carbapenems | 10.328 (6.740–15.825) | <0.001 |
| Exposure to BLBLIs | 2.776 (1.982–3.889) | <0.001 |

[a]CRKP, carbapenem-resistant *K. pneumoniae*; OR, odds ratio; CI, confidence interval; ICU, intensive care unit; BLBLIs, β-lactams–β-lactamase inhibitor combinations.

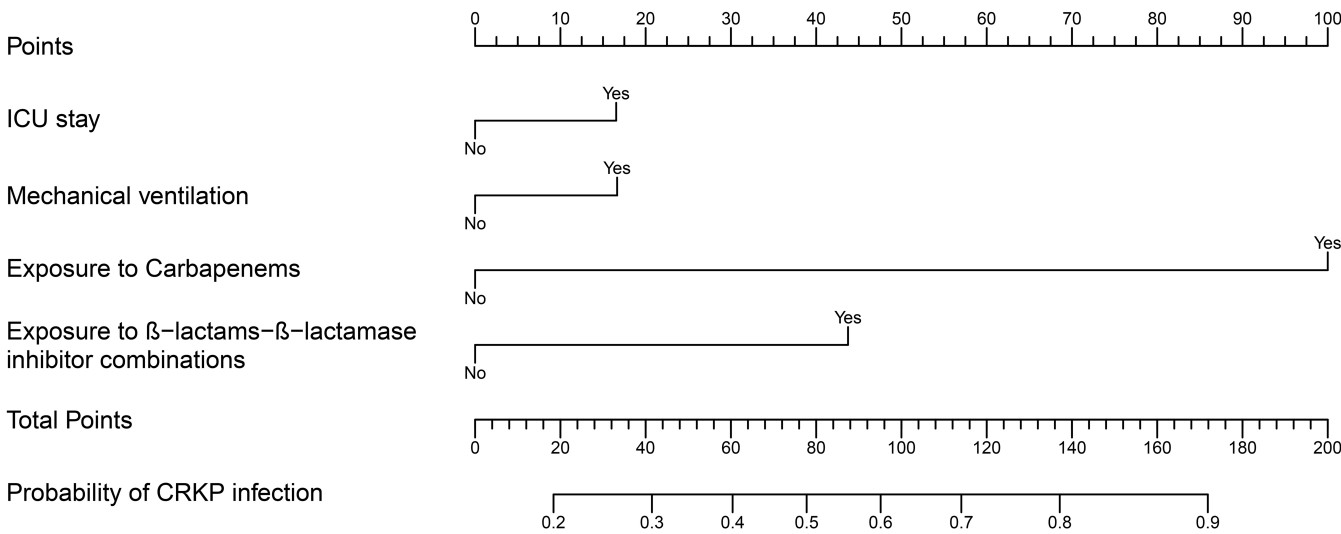

**FIG 2** The nomogram for predicting the risk of CRKP infection including ICU stay, mechanical ventilation, exposure to carbapenems, and exposure to BLBLIs. The probability of CRKP infection was calculated by adding the scores for each variable. CRKP, carbapenem-resistant *Klebsiella pneumoniae*; ICU, intensive care unit.

## Risk factors in the nomogram

This study retrospectively examined patient data concerning hospital-acquired KP infections at a tertiary general hospital. Through LASSO regression and multivariate logistic regression, we have identified four risk factors associated with CRKP, including prior carbapenem exposure, prior BLBLI exposure, prior ICU stay, and prior mechanical ventilation, which are common intervention factors during hospitalization. Based on the four independent predictors, we developed a nomogram for CRKP risk prediction, which proved to have strong discriminatory power, excellent calibration, and substantial clinical utility.

First, the results indicated that prior carbapenem exposure significantly increased the risk of CRKP infection. Previous studies have also highlighted the substantial impact of pre-infectious antibiotic exposure on CRKP infections (1, 19, 20). Prolonged use of antibiotics can lead to bacterial colonization by disrupting the balance of intestinal flora and selectively eliminating antibiotic-sensitive bacteria (21, 22). Additionally, research has shown that the widespread use of carbapenems can induce drug resistance mechanisms in KP through gene mutations and plasmid evolution (6).

Moreover, this study revealed that prior BLBLI exposure was also a risk factor for CRKP infection, which aligns with findings from earlier studies (20, 21). BLBLIs, such as ampicillin/sulbactam, piperacillin/tazobactams, and ceftazidime/avibactam (CZA), have been integrated into clinical practice since the 1980s to counteract β-lactamase-mediated resistance, with CZA being the most promising option for treating CRKP infections (23, 24). However, since 2015, the emergence of CZA-resistant KP strains has been reported (25, 26). A study conducted in Korea revealed a temporal correlation between exposure to BLBLIs and resistance to piperacillin/tazobactam in KP (27). More recent research has identified that KP may produce a variant KP carbapenemase (KPC) that is resistant to both CZA and carbapenem (28). Therefore, it is critical to balance the benefit-loss ratio associated with antibiotic use and to minimize the use of unnecessary antibiotics.

In line with previous studies, a history of ICU stays significantly increased the occurrence rate of CRKP infection (1, 20). The prevalence of multidrug-resistant KP (MRKP) colonization exceeded 50% among patients who remained in the ICU for more than 3 weeks (29). ICU patients often present with compromised immune systems and may require mechanical ventilation, indwelling catheters, and highly invasive procedures, all of which can increase the risk of CRKP infections (1, 19, 20). Furthermore, frequent interactions between clinicians and patients in confined environments can facilitate iatrogenic and airborne infections (30). Therefore, enhancing infection control practices

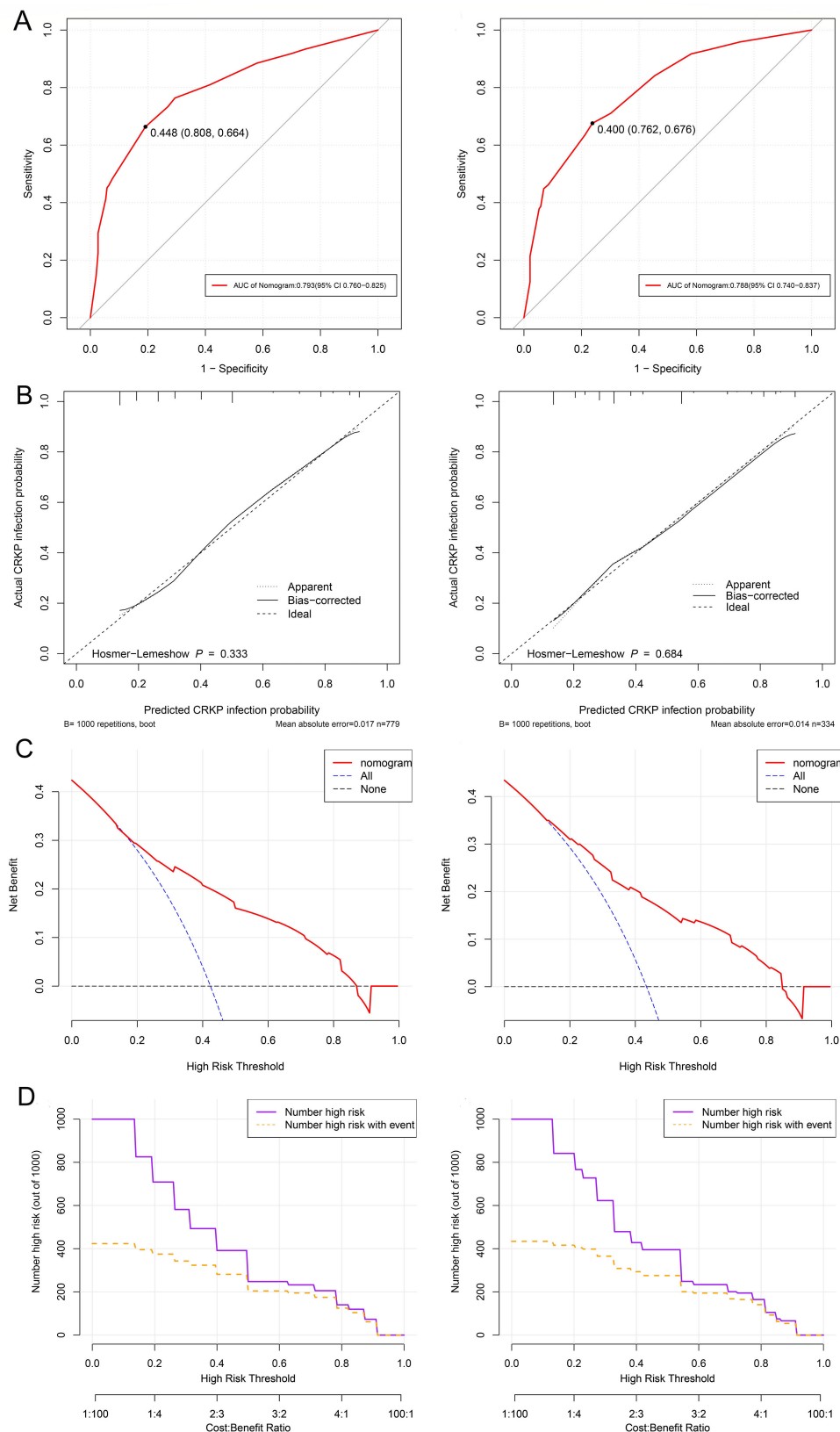

**FIG 3** Receiver operating characteristic (ROC) curve, calibration plots, decision curve analysis (DCA), and clinical impact curves (CIC). (A) ROC of the predictive model in the training and validation cohorts. (B) Calibration plots of the predictive

Fig 3 (Continued)

model in the training and validation cohorts. (C) DCA of the predictive model in the training and validation cohorts. (D) CIC of the predictive model in the training and validation cohorts. The black dashed lines indicate the net benefit when all hospitalized patients are assumed not to have developed CRKP infections and are not treated. The blue dashed lines indicate the net benefit when all hospitalized patients are assumed to have developed CRKP infections and have received treatment. The purple curves (number of high risk) indicate the number of people classified as positive (high risk) by the predictive model at each threshold probability. The yellow curves (number of high risk with event) show the number of true positives for each threshold probability. CRKP, carbapenem-resistant *Klebsiella pneumoniae*.

and raising awareness among medical staff, ensuring proper hand hygiene and environmental disinfection, and minimizing the duration of ICU stays are critical measures for preventing CRKP infections.

We found that mechanical ventilation was also an independent risk factor for CRKP infection in the present study, which was consistent with previous research findings (18, 20). Mechanical ventilation can facilitate the entry of KP into the respiratory tract due to respiratory muscle weakness and decreased mucociliary clearance (31). Additionally, KP has demonstrated a remarkable ability to colonize medical devices and instruments, allowing for transmission between patients (32). To prevent ventilator-associated infections, it is critical to enhance instrument disinfection, routinely monitor airway colonization, and assess the necessity for early extubation to reduce the duration of mechanical ventilation.

## Limitations

In contrast to prior investigations, our study encompasses a more expansive cohort screened for key risk factors associated with CRKP infection. A distinctive aspect of our work is the nuanced analysis of how prior carbapenem and BLBLIs exposure predisposes to such infections. The nomogram we have developed not only refines predictive accuracy for CRKP infections but also reveals the economic implications of prudent antibiotic use. Although our study includes a large population of patients from various departments within a tertiary general hospital, enhancing its representativeness, several limitations must be considered. First, despite the strong predictive power of our nomogram, its generalizability might be limited by the retrospective design and the single-center data source. This could expose our findings to selection bias, recall bias, and other confounding factors that could affect the validity of our results. Second, due to screening limitations, we were unable to assess the colonization status of CRKP at patient admission, potentially affecting the interpretation of infection risk. Finally, our study did not investigate the potential impact of combination antibiotic therapy on CRKP infection, leaving a gap in understanding how such treatment strategies may affect outcomes.

## Conclusions

In conclusion, this study meticulously analyzed the characteristics and clinical information of patients with nosocomial KP infections. Our findings revealed that prior carbapenems exposure, prior BLBLIs exposure, prior ICU stay, and prior mechanical ventilation were significant independent risk factors for CRKP infection. We have developed a nomogram predicated on these factors, which not only demonstrates robust discrimination and calibration capabilities but also offers a pragmatic framework for the prudent management of antimicrobial resources. This tool is designed to facilitate clinicians in devising timely preventative strategies and control measures against CRKP infections, thereby potentially enhancing patient outcomes and contributing to the broader effort to combat antibiotic resistance.

## ACKNOWLEDGMENTS

We sincerely thank the patients whose valuable data were vital to this research.

This study was supported by the Hubei Provincial Public Health Leading Talents Program (2021), and the authors acknowledge this financial support.

All authors contributed to the study and approved the final submitted version.

## AUTHOR AFFILIATIONS

[1]Department of Infectious Disease, Union Hospital, Tongji Medical College, Huazhong University of Science and Technology, Wuhan, China

[2]Department of Nosocomial Infection Management, Union Hospital, Tongji Medical College, Huazhong University of Science and Technology, Wuhan, China

## AUTHOR ORCIDs

Lijuan Xiong http://orcid.org/0000-0001-5643-4244

## FUNDING

| Funder | Grant(s) | Author(s) |
| --- | --- | --- |
| Health Commission of Hubei Province (湖北省卫生健康委员会) | the Hubei Provincial Public Health Leading Talents Program [2021] | Lijuan Xiong |

## AUTHOR CONTRIBUTIONS

Tingting Xu, Conceptualization, Data curation, Formal analysis, Investigation, Methodology, Project administration, Resources, Writing – original draft, Writing – review and editing | Yuxin Shi, Conceptualization, Data curation, Formal analysis, Investigation, Methodology, Project administration, Resources | Xiongjing Cao, Data curation, Formal analysis | Lijuan Xiong, Conceptualization, Data curation, Formal analysis, Funding acquisition, Investigation, Project administration, Resources

## DATA AVAILABILITY

The data will be made available in a suitable and accessible format, subject to any ethical or legal restrictions that may apply.

## ADDITIONAL FILES

The following material is available online.

Open Peer Review

**PEER REVIEW HISTORY (review-history.pdf).** An accounting of the reviewer comments and feedback.

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
