## [Reviewer comments · Microbiology Spectrum]

Microbiology Spectrum

Development and Validation of a Nomogram-Based Risk Prediction Model for Carbapenem-Resistant *Klebsiella pneumoniae* in Hospitalized Patients

Tingting Xu, Yuxin Shi, Xiongjing Cao, and Lijuan Xiong

Corresponding Author(s): Lijuan Xiong, Huazhong University of Science and Technology Tongji Medical College Union Hospital

Review Timeline:

Submission Date:	August 29, 2024
Editorial Decision:	September 29, 2024
Revision Received:	October 9, 2024
Accepted:	October 17, 2024

Editor: Gregory Wiedman

Reviewer(s): Disclosure of reviewer identity is with reference to reviewer comments included in decision letter(s). The following individuals involved in review of your submission have agreed to reveal their identity: Innocent Afeke (Reviewer #1); Rakshya Baral (Reviewer #3)

Transaction Report:

DOI: <https://doi.org/10.1128/spectrum.02170-24>

Re: Spectrum02170-24 (Assessing Risk Factors in Hospitalized Patients with Carbapenem-resistant *Klebsiella pneumoniae* Infection)

Dear Prof. Lijuan Xiong:

Thank you for the privilege of reviewing your work. Below you will find my comments, instructions from the Spectrum editorial office, and the reviewer comments.

Revision Guidelines

Sincerely,
Gregory Wiedman
Editor
Microbiology Spectrum

Reviewer #1 (Comments for the Author):

Abstract:

The abstract provides a clear overview of the study. The authors should include a brief note on the study's limitations, such as the retrospective design or reliance on data from a single hospital and suggest potential future research directions.

Introduction:

The introduction effectively introduces the problem of CRKP infections, providing background on *Klebsiella pneumoniae* and its resistance to carbapenems.

The authors should expand the introduction to include global data on CRKP infections, particularly in regions like Europe and the Americas, to emphasize the study's worldwide relevance.

Methods:

The authors should state under the study design that their study was a hospital-based retrospective study.

The CLSI unit for MIC is either $\mu\text{g/ml}$ or mg/L , not mg/ml , as quoted by the authors.

Justify the use of LASSO regression over other statistical methods, such as stepwise regression, and provide details on how missing or incomplete data was addressed. Additionally, explain why certain cutoff points were selected for the AUC and how they impact the model's practical utility.

Results:

Well presented

Discussion:

The authors should thoroughly discuss the study's limitations and potential biases, such as its retrospective design and the fact that data was only collected from a single tertiary hospital. If available, they should compare the nomogram's performance with other predictive tools for CRKP and discuss potential strategies for integrating it into everyday clinical practice. For example, "Although our nomogram demonstrated strong predictive power, its generalizability may be limited by the retrospective study design and the single-centre data source."

Conclusion: The authors should clearly state the nomogram's clinical applicability and suggest future research directions, such as validating the model in other healthcare settings or exploring its use in real-time clinical decision-making.

Reviewer #2 (Comments for the Author):

Reviewer' Comments to Author:

In this retrospective analysis, you explored risk factors of hospital-acquired CRKP by applying a non-standardised nomogram scoring system.

I due have some questions/suggestions to clarify few aspects:

MAJOR COMMENTS/QUESTIONS

1. Title should be revised based on what is the main focus of this study;

2. According the abstract and the data reported in the results and discussion, it is not clear what is the main purpose of this study: it is the exploration of risk factors of people that had infections with CRKP or the definition of a novel tool to for the prediction of CRKP infections.

Reviewer #3 (Comments for the Author):

The paper is well-organized, but adding specificity and clarity would enhance its readability. In the Abstract, specify the patient population and clarify the significance of the AUC values in validating the model. The Introduction effectively establishes the importance of CRKP, but further context on its public health impact, especially the implications of the WHO and CDC threat categorizations, would strengthen it. Additionally, clearly articulate how this study builds upon previous research and its contribution to the field. Ensure all sections are balanced in technicality, with explanations that can be understood by a broader clinical audience. Finally, a clearer breakdown of the Discussion section would enhance the overall flow of the paper.

**Assessing Risk Factors in Hospitalized Patients with Carbapenem-resistant**
***Klebsiella pneumoniae* Infection**
Tingting Xu ^{a,1}, Yuxin Shi ^{b,1}, Xiongjing Cao ^{b,1}, Lijuan Xiong ^{a,b,*}

4 ^a Department of Infectious Disease, Union Hospital, Tongji Medical College, Huazhong
University of Science and Technology, Wuhan, China

6 ^b Department of Nosocomial Infection Management, Union Hospital, Tongji Medical College,
Huazhong University of Science and Technology, Wuhan, China
¹ First authors, equally contributed.
*Corresponding author
Lijuan Xiong, Department of Infectious Disease & Department of Nosocomial Infection
Management, Union Hospital, Tongji Medical College, Huazhong University of Science and
Technology 1277 Jiefang Avenue, Wuhan 430022, China.
E-mail: lijuanxiong2016@126.com.
Tel: +8613971086426.
ORCID iD: <https://orcid.org/0000-0001-5643-4244>
**Abstract** Carbapenem-resistant *Klebsiella pneumoniae* (CRKP) poses one of the major
challenges in clinical anti-infective therapy worldwide. The aim of this study was to explore
the risk factors of hospital-acquired CRKP infections. We conducted a retrospective analysis
of 1,113 patients with hospital-acquired KP infections at a tertiary general hospital in Wuhan,
focusing on the associated risk factors, clinical features, and outcomes. All included patients
were randomly assigned to training (70%) and validation (30%) groups. In the training cohort,
independent risk factors for CRKP infection were identified using LASSO logistic regression
analysis and incorporated into a nomogram. The factors included in the nomogram were prior
carbapenems exposure, prior β -lactams- β -lactamase inhibitor combinations (BLBLIs)
exposure, prior intensive care unit (ICU) stay, and prior mechanical ventilation. We assessed
the discriminatory power and calibration of the nomogram in both the training and validation
groups, yielding areas under the curve (AUC) of 0.793 and 0.788, respectively. The p-values
for the Hosmer-Lemeshow test were 0.333 and 0.684, while the mean absolute errors for the
calibration curves were 0.017 and 0.014, respectively. Decision curve analysis (DCA) and
clinical impact curve (CIC) further demonstrated the high clinical utility of this nomogram.
**IMPORTANCE** We established a nomogram scoring system that incorporates four key risk
factors: prior carbapenems exposure, prior BLBLIs exposure, prior ICU stay, and prior
mechanical ventilation. This nomogram demonstrated strong discriminatory power, excellent
calibration, and significant clinical utility. This study highlights the critical risk factors
associated with hospital-acquired CRKP infections, providing valuable insights for clinicians
to identify high-risk patients.
**Keywords:** Carbapenem-resistance; *Klebsiella pneumoniae*; Nomogram; Risk factor
**Introduction**
*Klebsiella pneumoniae* (KP) is a member of intestinal flora and an opportunistic
pathogen capable of causing various types of infections, such as respiratory infections,
bloodstream infections, and intra-abdominal infections [1]. Additionally, KP also ranks among
the most common bacteria causing hospital-acquired infections [2]. In the early 1980s, KP
was found to be generally resistant to β -lactam antibiotics, and carbapenems, which have a
broader antimicrobial spectrum, became the main antibiotics used to treat KP [3-5]. However,
in recent years it has been found that carbapenem resistance mechanisms have emerged, and
the incidence of carbapenem-resistant KP (CRKP) is rapidly increasing [6]. In 2018 and 2019,
the World Health Organization (WHO) and the Centers for Disease Control and Prevention
(CDC) categorized CRKP as a top priority among antimicrobial-resistant bacteria and an
urgent threat to human health [7,8]. According to the China Antimicrobial Surveillance
Network (CHINET, <http://www.chinets.com/>), in the first half of 2023, the detected resistance
rates of KP to meropenem and imipenem in China were 30% and 29%, respectively [9]. Due
to high prevalence and mortality rates, CRKP has become a major public health hazard
worldwide [10].
Effective prevention and control of CRKP infections requires accurate risk assessment
tools. Previous studies have found that some underlying factors for CRKP include prolonged
hospitalization, ICU stay, KP colonization, receiving surgery, indwelling devices, previous
exposure to antibiotics, etc. [1,11-13] However, the results are generally based on
small-sample, single-center observational studies that have not been validated on a large scale
and have not generated prognostic models. The development of a predictive nomogram based
on comprehensive patient data and clinical parameters presents a promising avenue to identify
individuals at heightened risk of CRKP infection. This study aimed to retrospectively analyze
clinical information of adult patients with hospital-acquired KP infections from various
departments in a tertiary care hospital in China, establishing and validating a nomogram for
prediction of CRKP infection risk. Through this study, we seek to provide clinicians with a
reliable tool to enhance infection prevention strategies and guide tailored therapeutic
approaches.
**Methods**
*Patients and study design*

[revised manuscript text omitted]

to broader medical contexts, it is essential to acknowledge several inherent limitations. First,
as a single-center retrospective study, our research may be subject to selection bias, recall bias,
and other confounding factors that could affect the validity of the findings. Second, due to
screening limitations, we were unable to assess the colonization status of CRKP at the time of
patient admission, which may influence the interpretation of infection risk. Finally, our study
did not investigate the potential impact of combination antibiotic therapy on CRKP infection,
leaving a gap in understanding how such treatment strategies may affect outcomes.
**Conclusions**
In conclusion, this study analyzed the characteristics and clinical information of patients
with nosocomial KP infections and found that prior carbapenems exposure, prior BLBLIs
exposure, prior ICU stay, and prior mechanical ventilation were independent risk factors for
CRKP infection. A nomogram for CRKP prediction was constructed based on these factors
and presented excellent discrimination and calibration, which can help clinicians make
strategies in the timely prevention and control of CRKP infections, hoping to improve patient
prognoses.
**Acknowledgements**
We sincerely thank the patients whose valuable data were vital to this research.
*Author contributions.* TTX, YXS and XJC contributed equally to this work. TTX and
YXS collected and organized the data, TTX and XJC analyzed and interpreted the data results.
LJX and TTX drafted the manuscript. LJX conceived and designed the study. All authors
contributed to the study and approved the final submitted version.
*Funding.* This study was supported by the Hubei Provincial Public Health Leading
Talents Program (2021), and the authors acknowledge this financial support.
*Data Availability.* The data will be made available in a suitable and accessible format,
subject to any ethical or legal restrictions that may apply.
**Declarations**
*Conflict of Interests.* The authors declare that they have no competing financial
interests.
**Reference**
- [1] Lou T, Du X, Zhang P, et al. Risk factors for infection and mortality caused by
carbapenem-resistant *Klebsiella pneumoniae*: A large multicentre case-control and
cohort study. *J Infect.* 2022;84:637–47.
- [2] Lee CR, Lee JH, Park KS, et al. Antimicrobial Resistance of Hypervirulent *Klebsiella*
*pneumoniae*: Epidemiology, Hypervirulence-Associated Determinants, and Resistance
Mechanisms. *Front Cell Infect Microbiol.* 2017;7:483.
- [3] Canton R, Novais A, Valverde A, et al. Prevalence and spread of extended-spectrum
beta-lactamase-producing *Enterobacteriaceae* in Europe. *Clin Microbiol Infect.* 2008;14
Suppl 1:144–53.
- [4] Boix-Palop L, Xercavins M, Badia C, et al. Emerging extended-spectrum
beta-lactamase-producing *Klebsiella pneumoniae* causing community-onset urinary tract
infections: a case-control-control study. *Int J Antimicrob Agents.* 2017;50:197–202.
- [5] Livermore DM, Woodford N. The beta-lactamase threat in *Enterobacteriaceae*,
*Pseudomonas* and *Acinetobacter*. *Trends Microbiol.* 2006;14:413–20.
- [6] Park SO, Liu J, Furuya EY, Larson EL. Carbapenem-Resistant *Klebsiella pneumoniae*
Infection in Three New York City Hospitals Trended Downwards From 2006 to 2014.
*Open Forum Infect Dis.* 2016;3:ofw222.
- [7] Tacconelli E, Carrara E, Savoldi A, et al. Discovery, research, and development of new
antibiotics: the WHO priority list of antibiotic-resistant bacteria and tuberculosis. *Lancet*
*Infect Dis.* 2018;18:318–27.
- [8] Centers for Disease Control and Prevention (U.S.). Antibiotic resistance threats in the
- United States, 2019. <https://stacks.cdc.gov/view/cdc/82532>.
- Accessed 15 October 2023.
- [9] CHINET. China antimicrobial resistance surveillance system report.
- <http://www.chinets.com/>.
- Accessed 15 October 2023.
- [10] Cantón R, Akóva M, Carmeli Y, et al. Rapid evolution and spread of carbapenemases
- among Enterobacteriaceae in Europe. *Clin Microbiol Infect.* 2012;18:413–31.
- [11] Xiao T, Zhu Y, Zhang S, et al. A Retrospective Analysis of Risk Factors and Outcomes
- of Carbapenem-Resistant *Klebsiella pneumoniae* Bacteremia in Nontransplant Patients. *J*
- *Infect Dis.* 2020;221:S174–S83.
- [12] Chang H, Wei J, Zhou W, et al. Risk factors and mortality for patients with Bloodstream
- infections of *Klebsiella pneumoniae* during 2014-2018: Clinical impact of carbapenem
- resistance in a large tertiary hospital of China. *J Infect Public Health.* 2020;13:784–90.
- [13] Zhang G, Zhang M, Sun F, et al. Epidemiology, mortality and risk factors for patients
- with *K. pneumoniae* bloodstream infections: Clinical impact of carbapenem resistance in
- a tertiary university teaching hospital of Beijing. *J Infect Public Health.* 2020;13:1710–4.
- [14] Kollef MH, Torres A, Shorr AF, Martin-Loeches I, Micek ST. Nosocomial Infection. *Crit*
- *Care Med.* 2021;49:169–87.
- [15] Osme SF, Almeida APS, Lemes MF, et al. Costs of healthcare-associated infections to
- the Brazilian public Unified Health System in a tertiary-care teaching hospital: a
- matched case-control study. *J Hosp Infect.* 2020;106:303–10.
- [16] Wyres KL, Holt KE. *Klebsiella pneumoniae* as a key trafficker of drug resistance genes
from environmental to clinically important bacteria. *Curr Opin Microbiol.* 2018;45:131–
9.
[17] Wang Z, Qin RR, Huang L, Sun LY. Risk Factors for Carbapenem-resistant *Klebsiella*
*pneumoniae* Infection and Mortality of *Klebsiella pneumoniae* Infection. *Chin Med J*
(Engl). 2018;131:56–62.
[18] Li M, Yang S, Yao H, Liu Y, Du M. Retrospective Analysis of Epidemiology, Risk
Factors, and Outcomes of Health Care-Acquired Carbapenem-Resistant *Klebsiella*
*pneumoniae* Bacteremia in a Chinese Tertiary Hospital, 2010-2019. *Infect Dis Ther.*
2023;12:473–85.
[19] Ding D, Wang B, Zhang X, et al. The spread of antibiotic resistance to humans and
potential protection strategies. *Ecotoxicol Environ Saf.* 2023;254:114734.
[20] Qin X, Wu S, Hao M, et al. The Colonization of Carbapenem-Resistant *Klebsiella*
*pneumoniae*: Epidemiology, Resistance Mechanisms, and Risk Factors in Patients
Admitted to Intensive Care Units in China. *J Infect Dis.* 2020;221:S206–S14.
[21] Drawz SM, Bonomo RA. Three decades of beta-lactamase inhibitors. *Clin Microbiol*
*Rev.* 2010;23:160–201.
[22] Liu X, Chu Y, Yue H, Huang X, Zhou G. Risk factors for and clinical outcomes of
ceftazidime-avibactam-resistant carbapenem-resistant *Klebsiella pneumoniae*
nosocomial infections: a single-center retrospective study. *Infection.* 2022;50:1147–54.
[23] Findlay J, Poirel L, Bouvier M, Gaia V, Nordmann P. Resistance to
ceftazidime-avibactam in a KPC-2-producing *Klebsiella pneumoniae* caused by the
extended-spectrum beta-lactamase VEB-25. *Eur J Clin Microbiol Infect Dis.*
2023;42:639–44.
[24] Humphries RM, Yang S, Hemarajata P, et al. First Report of Ceftazidime-Avibactam
Resistance in a KPC-3-Expressing *Klebsiella pneumoniae* Isolate. *Antimicrob Agents*
*Chemother.* 2015;59:6605–7.
[25] Ryu S, Klein EY, Chun BC. Temporal association between antibiotic use and resistance
in *Klebsiella pneumoniae* at a tertiary care hospital. *Antimicrob Resist Infect Control.*
2018;7:83.
[26] Di Pilato V, Principe L, Andriani L, et al. Deciphering variable resistance to novel
carbapenem-based beta-lactamase inhibitor combinations in a multi-clonal outbreak
caused by *Klebsiella pneumoniae* carbapenemase (KPC)-producing *Klebsiella*
*pneumoniae* resistant to ceftazidime/avibactam. *Clin Microbiol Infect.* 2023;29:537 e1–
e8.
[27] Ruiz J, Gordon M, Villarreal E, et al. Influence of antibiotic pressure on multi-drug
resistant *Klebsiella pneumoniae* colonisation in critically ill patients. *Antimicrob Resist*
*Infect Control.* 2019;8:38.
[28] Maina JW, Onyambu FG, Kibet PS, Musyoki AM. Multidrug-resistant Gram-negative
bacterial infections and associated factors in a Kenyan intensive care unit: a
cross-sectional study. *Ann Clin Microbiol Antimicrob.* 2023;22:85.
[29] Papazian L, Klompas M, Luyt CE. Ventilator-associated pneumonia in adults: a narrative
review. *Intensive Care Med.* 2020;46:888–906.
[30] Bonten MJ, Bergmans DC, Ambergen AW, et al. Risk factors for pneumonia, and
colonization of respiratory tract and stomach in mechanically ventilated ICU patients.
Am J Respir Crit Care Med. 1996;154:1339–46.
**Figure Legends**
**Fig. 1 (A)** LASSO coefficient profiles of the 35 risk factors. Each curve in the figure presents
the change of each variable in coefficient. **(B)** 10-fold cross-validation fitting and then
selecting the mode.
**Fig. 2** The nomogram for predicting the risk of CRKP infection including ICU stay,
Mechanical ventilation, Exposure to Carbapenems, Exposure to BLBLIs. The probability of
CRKP infection was calculated by adding the scores for each variable. CRKP:
carbapenem-resistant *Klebsiella pneumoniae*. ICU: intensive care unit.
**Fig. 3** Receiver operating characteristic (ROC) curve, Calibration plots, Decision curve
analysis (DCA) and Clinical impact curves (CIC). **(A)** ROC of the predictive model in the
training and validation cohort. **(B)** Calibration plots of the predictive model in the training and
validation cohort. **(C)** DCA of the predictive model in the training and validation cohort. **(D)**
CIC of the predictive model in the training and validation cohort. And the black dashed line
indicates the net benefit when all hospitalized patients are assumed not to have developed
CRKP infections and are not treated. The blue dashed line indicates the net benefit when all
hospitalized patients are assumed to have developed CRKP infections and have received
treatment. The purple curve (number of high risk) indicates the number of people classified as
positive (high risk) by the predictive model at each threshold probability. The yellow curve
(number high) risk with event is the number of true positives for each threshold probability.
CRKP: carbapenem-resistant *Klebsiella pneumoniae*.

Author's response to the editor and reviewers:

Dear Prof. Wiedman and reviewers,

We would like to thank the reviewers and the editor for the positive and constructive comments and suggestions. We have substantially revised our manuscript after reading the comments provided by the reviewers, and found these comments are very helpful. Below we respond to each of the referees' comments in turn.

1. Respond to Reviewer #1's comments

Reviewer #1 (Comments for the Author):

Abstract:

The abstract provides a clear overview of the study. The authors should include a brief note on the study's limitations, such as the retrospective design or reliance on data from a single hospital and suggest potential future research directions.

Response: Thank you for your comment. In response to this feedback, we have revised the abstract to include the following:

“Our retrospective analysis identified key risk factors for CRKP infection and developed a nomogram that could effectively predict CRKP infections in hospitalized patients. While the single-center nature of this study limits generalizability, the nomogram provides a foundation for future prospective, multicenter validations.” (Page 2, lines 28 to 32)

Introduction:

The introduction effectively introduces the problem of CRKP infections, providing background on *Klebsiella pneumoniae* and its resistance to carbapenems.

The authors should expand the introduction to include global data on CRKP infections, particularly in regions like Europe and the Americas, to emphasize the study's worldwide relevance.

Response: Thank you for your comment. Regarding the global relevance of CRKP infections, we have added the following to the revised manuscript:

“This categorization underscores the alarming rates of carbapenem resistance observed in KP, the species most affected within the Enterobacterales order. Notably, resistance rates have reached highs in certain regions, such as Greece at 67%, Iran at 65%, Russia at 64%, India at 57%, Saudi Arabia at 50%, Peru at 45%, Italy at 33%, Argentina at 26%, and Brazil at 24%. [World Health Organization (WHO) GLASS report: Early implementation. 2020.] [Nature communications, 15(1), 5092.]” (Page 4, lines 53 to 57)

Methods:

The authors should state under the study design that their study was a hospital-based retrospective study.

Response: Thank you for your helpful suggestions. We have added clarification in the study design section that this research is a hospital-based retrospective study, as detailed below:

“This retrospective study was conducted on patients with nosocomial infections of KP between January 1, 2020, and April 30, 2023, at Union Hospital of Huazhong University of Science and Technology.” (Page 6, lines 76 to 78)

The CLSI unit for MIC is either $\mu\text{g/ml}$ or mg/L , not mg/ml , as quoted by the authors.

Response: Thank you for your helpful suggestions. We have corrected the unit for MIC to " $\mu\text{g/ml}$ " in the revised manuscript. (Page 7, lines 100)

Justify the use of LASSO regression over other statistical methods, such as stepwise regression, and provide details on how missing or incomplete data was addressed.

Response: Thank you for your insightful comments and questions. We have taken the time to address each point in detail as follows:

We selected LASSO regression for our analysis due to its superior handling of multicollinearity among the predictor variables. Unlike other methods, LASSO is capable of performing both variable selection and regularization, which helps to reduce the impact of multicollinearity on the model's performance. By penalizing the absolute size of the regression coefficients, LASSO shrinks some coefficients to exactly zero, effectively selecting a simpler model that is less likely to overfit the data.

Our study's inclusion criteria were deliberately designed to exclude patients with missing data. (Page 6, lines 79 to 80). This approach was taken to ensure the integrity and reliability of our analysis, as missing data can introduce bias and diminish the statistical power of a study.

Additionally, explain why certain cutoff points were selected for the AUC and how they impact the model's practical utility.

Response: Thank you for your comment. The optimal cutoff values were determined by identifying the points on the ROC curve that are closest to the upper left corner, representing the best balance between sensitivity and specificity. These points were selected as they maximize the Youden's Index (sensitivity + specificity - 1), which is a standard method for choosing the best cutoff in diagnostic tests.

Results:

Well presented

Response: Thank you for your comment.

Discussion:

The authors should thoroughly discuss the study's limitations and potential biases, such as its retrospective design and the fact that data was only collected from a single tertiary hospital. If

available, they should compare the nomogram's performance with other predictive tools for CRKP and discuss potential strategies for integrating it into everyday clinical practice. For example, "Although our nomogram demonstrated strong predictive power, its generalizability may be limited by the retrospective study design and the single-centre data source.

Response: Thank you for your helpful suggestions. We have taken them into consideration and have thoroughly discussed the limitations and biases of this study in detail in the revised manuscript, which is revised as follows:

“While our study includes a large population of patients from various departments within a tertiary general hospital, enhancing its representativeness, several limitations must be considered. First, despite the strong predictive power of our nomogram, its generalizability might be limited by the retrospective design and the single-center data source. This could expose our findings to selection bias, recall bias, and other confounding factors that could affect the validity of our results. Second, due to screening limitations, we were unable to assess the colonization status of CRKP at patient admission, potentially affecting the interpretation of infection risk.” (Page 13, lines 217 to 224)

We appreciate the reviewer's suggestion to compare our nomogram's performance with other predictive tools for CRKP. But it is important to note that our nomogram incorporates distinct variables not previously considered in other models, which limits the direct comparability. Our model's novelty lies in its inclusion of prior carbapenem exposure and prior BLBLIs exposure, which addresses a gap in the existing predictive tools and provides a more comprehensive assessment of CRKP risk. We describe potential strategies for integrating this nomogram into daily clinical practice as follows:

“In contrast to prior investigations, our study encompasses a more expansive cohort screened for key risk factors associated with CRKP infection. A distinctive aspect of our work is the nuanced analysis of how prior carbapenem and BLBLIs exposure predisposes to such infections. The nomogram we've developed not only refines predictive accuracy for CRKP infections but also reveals the economic implications of prudent antibiotic use.” (Page 13, lines 213 to 217)

Conclusion: The authors should clearly state the nomogram's clinical applicability and suggest future research directions, such as validating the model in other healthcare settings or exploring its use in real-time clinical decision-making.

Response: Thank you for your comment. we have added the following to the revised manuscript: “We have developed a nomogram predicated on these factors, which not only demonstrates robust discrimination and calibration capabilities but also offers a pragmatic framework for the prudent management of antimicrobial resources. This tool is designed to facilitate clinicians in devising timely preventative strategies and control measures against CRKP infections, thereby potentially enhancing patient outcomes and contributing to the broader effort to combat antibiotic resistance.” (Page 15, lines 231 to 236)

2. Respond to Reviewer #2's comments

Reviewer #2 (Comments for the Author):

Reviewer' Comments to Author:

In this retrospective analysis, you explored risk factors of hospital-acquired CRKP by applying a non-standardised nomogram scoring system.

I due have some questions/suggestions to clarify few aspects:

MAJOR COMMENTS/QUESTIONS

1. Title should be revised based on what is the main focus of this study;

Response: Thank you for your insightful comments and suggestions. We have revised the title of our manuscript to more accurately reflect the dual focus of our study, which is both the exploration of risk factors and the development and validation of a predictive nomogram for CRKP infections. Our revised title is as follows:

“Development and Validation of a Nomogram-Based Risk Prediction Model for Carbapenem-Resistant *Klebsiella pneumoniae* in Hospitalized Patients”

2. According the abstract and the data reported in the results and discussion, it is not clear what is the main purpose of this study: it is the exploration of risk factors of people that had infections with CRKP or the definition of a novel tool to for the prediction of CRKP infections.

Response: Thank you for your insightful comments and suggestions. We made clear in the abstract the objective of our study: to identify the predominant risk factors for hospital-acquired CRKP infection and to develop a corresponding nomogram for predicting CRKP risk. A nomogram is a graphical tool used to predict outcomes or make decisions based on a set of variables by visually plotting their values on a calibrated chart. We have revised the abstract to include the following:

“Independent risk factors were identified using LASSO logistic regression and incorporated into a predictive nomogram” (Page 2, lines 19 to 21)

3. Respond to Reviewer #3's comments

Reviewer #3 (Comments for the Author):

The paper is well-organized, but adding specificity and clarity would enhance its readability. In the Abstract, specify the patient population and clarify the significance of the AUC values in validating the model.

Response: Thank you for your insightful comments and suggestions. We have revised the abstract to include the following:

“This retrospective cohort study at a tertiary general hospital in Wuhan aimed to identify risk

factors for hospital-acquired CRKP infections among 1,113 patients. All participants were aged 18 years and above, and had confirmed positive cultures for KP isolated within 48 hours post-hospitalization. [Independent...ventilation]. The areas under the receiver operating characteristic curve (AUC) for the nomogram were 0.793 in the training group (70% of patients) and 0.788 in the validation group (30% of patients), demonstrating its discriminatory power and predictive accuracy.” (Page 2, lines 17 to 27)

The Introduction effectively establishes the importance of CRKP, but further context on its public health impact, especially the implications of the WHO and CDC threat categorizations, would strengthen it.

Response: Thank you for your insightful comments and suggestions. We have revised the Introduction to include the following:

“In 2018 and 2019, the World Health Organization (WHO) and the Centers for Disease Control and Prevention (CDC) prioritized antibiotic-resistant bacteria based on criteria including mortality, healthcare impact, resistance prevalence, and transmissibility, dividing them into critical, high, and medium priority tiers at the 33rd percentile threshold. which CRKP was categorized as a critical priority and an urgent threat to human health. This categorization underscores the alarming rates of carbapenem resistance observed in KP, the species most affected within the Enterobacterales order.” (Page 4, lines 49 to 56)

Additionally, clearly articulate how this study builds upon previous research and its contribution to the field. Ensure all sections are balanced in technicality, with explanations that can be understood by a broader clinical audience.

Response: Thank you for your insightful comments and suggestions. We have revised the Introduction to include the following:

“Effective prevention and control of CRKP infections requires accurate risk assessment tools. While existing research has identified factors such as prolonged hospitalization, ICU stay, KP colonization, receiving surgery, indwelling devices, as well as prior antibiotic exposure as potential predictors of CRKP [1, 13-15], these findings are often derived from small-sample, single-center observational studies that have not been validated on a large scale and have not generated prognostic models. Building on this foundation, our study leverages a large retrospective cohort from a tertiary care hospital in China to develop and validate a predictive nomogram for CRKP infection risk. This nomogram, grounded in extensive patient data and clinical parameters, offers a significant advancement by enhancing the accuracy of risk prediction. Our research contributes to the field by providing a practical tool that can help clinicians better anticipate CRKP infections. This nomogram not only aids in refining infection prevention strategies but also guides personalized treatment plans.” (Page 5, lines 63 to 74)

Finally, a clearer breakdown of the Discussion section would enhance the overall flow of the paper.

Response: Thank you for your insightful comments and suggestions. We have introduced clear

subheadings within the Discussion section to enhance readability and organization. The new subheadings are: "Clinical relevance", "Risk Factors in the Nomogram" and "Limitations." (Page 11–13)

We have endeavored to address all concerns raised by the reviewers to the best of our ability. We believe these revisions have significantly improved the manuscript's clarity and academic rigor, and enhanced its clinical relevance. We appreciate the reviewers' and your insightful suggestions.

Thank you for your comments and suggestions. We look forward to your response.

Best regards,

Lijuan Xiong, PhD

Re: Spectrum02170-24R1 (Development and Validation of a Nomogram-Based Risk Prediction Model for Carbapenem-Resistant *Klebsiella pneumoniae* in Hospitalized Patients)

Dear Prof. Lijuan Xiong:

Your manuscript has been accepted, and I am forwarding it to the ASM production staff for publication. Your paper will first be checked to make sure all elements meet the technical requirements. ASM staff will contact you if anything needs to be revised before copyediting and production can begin. Otherwise, you will be notified when your proofs are ready to be viewed.

Sincerely,
Gregory Wiedman
Editor
Microbiology Spectrum

Reviewer #1 (Comments for the Author):

The authors have adequately addressed my concerns.